# A Filtering Approach to Stochastic Variational Inference

**Neil M.T. Houlsby** *
Google Research
Zurich, Switzerland
neilhoulsby@google.com

**David M. Blei**
Department of Statistics
Department of Computer Science
Colombia University
david.blei@colombia.edu

## Abstract

Stochastic variational inference (SVI) uses stochastic optimization to scale up Bayesian computation to massive data. We present an alternative perspective on SVI as approximate parallel coordinate ascent. SVI trades-off bias and variance to step close to the unknown true coordinate optimum given by batch variational Bayes (VB). We define a model to automate this process. The model infers the location of the next VB optimum from a sequence of noisy realizations. As a consequence of this construction, we update the variational parameters using Bayes rule, rather than a hand-crafted optimization schedule. When our model is a Kalman filter this procedure can recover the original SVI algorithm and SVI with adaptive steps. We may also encode additional assumptions in the model, such as heavy-tailed noise. By doing so, our algorithm outperforms the original SVI schedule and a state-of-the-art adaptive SVI algorithm in two diverse domains.

## 1 Introduction

Stochastic variational inference (SVI) is a powerful method for scaling up Bayesian computation to massive data sets [1]. It has been successfully used in many settings, including topic models [2], probabilistic matrix factorization [3], statistical network analysis [4, 5], and Gaussian processes [6]. SVI uses stochastic optimization to fit a variational distribution, following cheap-to-compute noisy natural gradients that arise from repeatedly subsampling the data. The algorithm follows these gradients with a decreasing step size [7]. One nuisance, as for all stochastic optimization techniques, is setting the step size schedule.

In this paper we develop *variational filtering*, an alternative perspective of stochastic variational inference. We show that this perspective leads naturally to a tracking algorithm—one based on a Kalman filter—that effectively adapts the step size to the idiosyncrasies of data subsampling. Without any tuning, variational filtering performs as well or better than the best constant learning rate chosen in retrospect. Further, it outperforms both the original SVI algorithm and SVI with adaptive learning rates [8].

In more detail, variational inference optimizes a high-dimensional variational parameter $\lambda$ to find a distribution that approximates an intractable posterior. A concept that is important in SVI is the *parallel coordinate update*. This refers to setting each dimension of $\lambda$ to its coordinate optimum, but where these coordinates are computed parallel. We denote the resulting updated parameters $\lambda^{\text{VB}}$.

With this definition we have a new perspective on SVI. At each iteration it attempts to reach its parallel coordinate update, but one estimated from a randomly sampled data point. (The true coordinate update requires iterating over all of the data.) Specifically, SVI iteratively updates an estimate of $\lambda$

as follows,

$$\lambda_t = (1 - \rho_t)\lambda_{t-1} + \rho_t \hat{\lambda}_t, \tag{1}$$

where $\hat{\lambda}_t$ is a random variable whose expectation is $\lambda_t^{\text{VB}}$ and $\rho_t$ is the learning rate. The original paper on SVI points out that this iteration works because $\lambda_t^{\text{VB}} - \lambda_t$ is the natural gradient of the variational objective, and so Eq 1 is a noisy gradient update. But we can also see the iteration as a noisy attempt to reach the parallel coordinate optimum $\lambda_t^{\text{VB}}$. While $\hat{\lambda}$ is an unbiased estimate of this quantity, we will show that Eq 1 uses a biased estimate but with reduced variance.

This new perspective opens the door to other ways of updating $\lambda_t$ based on the noisy estimates of $\lambda_t^{\text{VB}}$. In particular, we use a Kalman filter to track the progress of $\lambda_t$ based on the sequence of noisy coordinate updates. This gives us a 'meta-model' about the optimal parameter, which we now estimate through efficient inference. We show that one setting of the Kalman filter corresponds to SVI; another corresponds to SVI with adaptive learning rates; and others, like using a t-distribution in place of a Gaussian, account better for noise than any previous methods.

## 2  Variational Filtering

We first introduce stochastic variational inference (SVI) as approximate parallel coordinate ascent. We use this view to present variational filtering, a model-based approach to variational optimization that observes noisy parallel coordinate optima and seeks to infer the true VB optimum. We instantiate this method with a Kalman filter, discuss relationships to other optimization schedules, and extend the model to handle real-world SVI problems.

**Stochastic Variational Inference**    Given data $x_{1:N}$, we want to infer the posterior distribution over model parameters $\theta$, $p(\theta|x_{1:N})$. For most interesting models exact inference is intractable and we must use approximations. Variational Bayes (VB) formulates approximate inference as a batch optimization problem. The intractable posterior distribution $p(\theta|x_{1:N})$ is approximated by a simpler distribution $q(\theta; \lambda)$ where $\lambda$ are the *variational* parameters of $q$.[1] These parameters are adjusted to maximize a lower bound on the model evidence (the ELBO),

$$\mathcal{L}(\lambda) = \sum_{i=1}^{N} \mathbb{E}_q[\log p(x_i|\theta)] + \mathbb{E}_q[\log p(\theta)] - \mathbb{E}_q[\log q(\theta)]. \tag{2}$$

Maximizing Eq 2 is equivalent to minimizing the KL divergence between the exact and approximate posterior, $\text{KL}[q||p]$. Successive optima of the ELBO often have closed-form [1], so to maximize Eq 2 VB can perform successive parallel coordinate updates on the elements in $\lambda$, $\lambda_{t+1} = \lambda_t^{\text{VB}}$.

Unfortunately, the sum over all $N$ datapoints in Eq 2 means that $\lambda_t^{\text{VB}}$ is too expensive on large datasets. SVI avoids this difficulty by sampling a single datapoint (or a mini-batch) and optimizing a cheap, noisy estimate of the ELBO $\hat{\mathcal{L}}(\lambda)$. The optimum of $\hat{\mathcal{L}}(\lambda)$ is denoted $\hat{\lambda}_t$,

$$\hat{\mathcal{L}}(\lambda) = N\mathbb{E}_q[\log p(x_i|\theta)] + \mathbb{E}_q[\log p(\theta)] - \mathbb{E}_q[\log q(\theta)], \tag{3}$$

$$\hat{\lambda} := \underset{\lambda}{\operatorname{argmax}} \hat{\mathcal{L}}(\lambda) = \mathbb{E}_q[N \log p(x_i|\theta) + \log p(\theta)]. \tag{4}$$

The constant $N$ in Eq 4 ensures the noisy parallel coordinate optimum is unbiased with respect to the full VB optimum, $\mathbb{E}[\hat{\lambda}_t] = \lambda_t^{\text{VB}}$. After computing $\hat{\lambda}_t$, SVI updates the parameters using Eq 1. This corresponds to using natural gradients [9] to perform stochastic gradient ascent on the ELBO.

We present an alternative perspective on Eq 1. SVI may be viewed as an attempt to reach the true parallel coordinate optimum $\lambda_t^{\text{VB}}$ using the noisy estimate $\hat{\lambda}_t$. The observation $\hat{\lambda}_t$ is an unbiased estimator of $\lambda_t^{\text{VB}}$ with variance $\text{Var}[\hat{\lambda}_t]$. The variance may be large, so SVI makes a bias/variance trade-off to reduce the overall error. The bias and variance in $\lambda_t$ computed using SVI (Eq 1) are

$$\mathbb{E}[\lambda_t - \lambda_t^{\text{VB}}] = (1 - \rho_t)(\lambda_{t-1} - \lambda_t^{\text{VB}}), \quad \text{Var}[\lambda_t] = \rho_t^2 \text{Var}[\hat{\lambda}_t], \tag{5}$$

respectively. Decreasing the step size reduces the variance but increases the bias. However, as the algorithm converges, the bias decreases as the VB optima fall closer to the current parameters. Thus,

$\lambda_{t-1} - \lambda_t^{\text{VB}}$ tends to zero and as optimization progresses, $\rho_t$ should decay. This reduces the variance given the same level of bias.

Indeed, most stochastic optimization schedules decay the step size, including the Robbins-Monro schedule [7] used in SVI. Different schedules yield different bias/variance trade-offs, but the trade-off is heuristic and these schedules often require hand tuning. Instead we use a model to infer the location of $\lambda_t^{\text{VB}}$ from the observations, and use Bayes rule to determine the optimal step size.

**Probabilistic Filtering for SVI**    We described our view of SVI as approximate parallel coordinate ascent. With this perspective, we can define a model to infer $\lambda_t^{\text{VB}}$. We have three sets of variables: $\lambda_t$ are the current parameters of the approximate posterior $q(\theta; \lambda_t)$; $\lambda_t^{\text{VB}}$ is a hidden variable corresponding to the VB coordinate update at the current time step; and $\hat{\lambda}_t$ is an unbiased, but noisy observation of $\lambda_t^{\text{VB}}$.

We specify a model that observes the sequence of noisy coordinate optima $\hat{\lambda}_{1:t}$, and we use it to compute a distribution over the full VB update $p(\lambda_t^{\text{VB}}|\hat{\lambda}_{1:t})$. When making a parallel coordinate update at time $t$ we move to the best estimate of the VB optimum under the model, $\lambda_t = \mathbb{E}[\lambda_t^{\text{VB}}|\hat{\lambda}_{1:t}]$. Using this approach we i) avoid the need to tune the step size because Bayes rule determines how the posterior mean moves at each iteration; ii) can use a Kalman filter to recover particular static and adaptive step size algorithms; and iii) can add extra modelling assumptions to vary the step size schedule in useful ways.

In variational inference, our 'target' is $\lambda_t^{\text{VB}}$. It moves because the parameters of approximate posterior $\lambda_t$ change as optimization progresses. Therefore, we use a dynamic tracking model, the Kalman filter [10]. We compute the posterior over next VB optimum given previous observations, $p(\lambda_t^{\text{VB}}|\hat{\lambda}_{1:t})$. In tracking, this is called filtering, so we call our method variational filtering (VF).[2] At each time $t$, VF has a current set of model parameters $\lambda_{t-1}$ and takes these steps.

1. Sample a datapoint $x_t$.
2. Compute the noisy estimate of the coordinate update $\hat{\lambda}_t$ using Eq 3.
3. Run Kalman filtering to compute the posterior over the VB optimum, $p(\lambda_t^{\text{VB}}|\hat{\lambda}_{1:t})$.
4. Update the parameters to the posterior mean $\lambda_t = \mathbb{E}[\lambda_t^{\text{VB}}|\hat{\lambda}_{1:t}]$ and repeat.

Variational filtering uses the entire history of observations, encoded by the posterior, to infer the location of the VB update. Standard optimization schedules use only the current parameters $\lambda_t$ to regularize the noisy coordinate update, and these methods require tuning to balance bias and variance in the update. In our setting, Bayes rule automatically makes this trade-off.

To illustrate this perspective we consider a small problem. We fit a variational distribution for latent Dirichlet allocation on a small corpus of 2.5k documents from the ArXiv. For this problem we can compute the full parallel coordinate update and thus compute the tracking error $||\lambda_t^{\text{VB}} - \lambda_t||_2^2$ and the observation noise $||\lambda_t^{\text{VB}} - \hat{\lambda}_t||_2^2$ for various algorithms. We emphasize that $\hat{\lambda}_t$ is unbiased, and so the observation noise is completely due to variance. A reduction in tracking error indicates an advantage to incurring bias for a reduction in variance.

We compared variational filtering (Alg. 1) to the original Robbins-Monro schedule used in SVI [1], and a large constant step size of $0.5$. The same sequence of random documents was handed to each algorithm. Figs. 1 (a-c) show the tracking error of each algorithm. The large constant step size yields large error due to high variance, see Eq 5. The SVI updates are too small and the bias dominates. Here, the bias is even larger than the variance in the noisy observations during early stages, but it decays as the term $(\lambda_t - \lambda_{t-1}^{\text{VB}})$ in Eq 5 slowly decreases. The variational filter automatically balances bias and variance, yielding the smallest tracking error. As a result of following the VB optima more closely, the variational filter achieves larger values of the ELBO, shown in Fig. 1 (d).

## 3   Kalman Variational Filter

We now detail our Kalman filter for SVI. Then we discuss different settings of the parameters and estimating these online. Finally, we extend the filter to handle heavy-tailed noise.

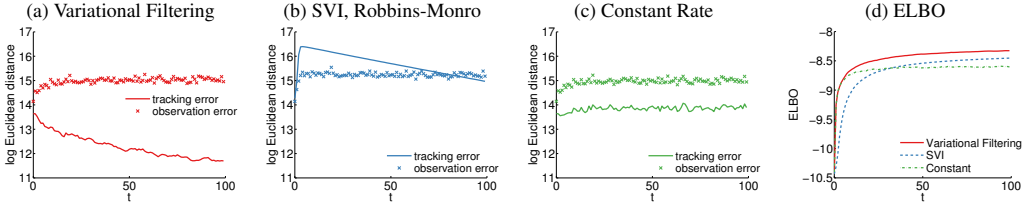

Figure 1: (a-c) Curves show the error in tracking the VB update. Markers depict the error in the noisy observations $\hat{\lambda}_t$ to the VB update. (d) Evolution of the ELBO computed on the entire dataset.

The Gaussian Kalman filter (KF) is attractive because inference is tractable and, in SVI, computational time is the limiting factor, not the rate of data acquisition. The model is specified as

$$p(\lambda_{t+1}^{\text{VB}}|\lambda_t^{\text{VB}}) = \mathcal{N}(\lambda_t^{\text{VB}}, Q), \qquad p(\hat{\lambda}_t|\lambda_t^{\text{VB}}) = \mathcal{N}(\lambda_t^{\text{VB}}, R), \tag{6}$$

where $R$ models the variance in the noisy coordinate updates and $Q$ models how far the VB optima move at each iteration. The observation noise has zero mean because the noisy updates are unbiased. We assume no systematic parameter drift, so $\mathbb{E}[\lambda_{t+1}^{\text{VB}}] = \lambda_t^{\text{VB}}$. Filtering in this linear-Gaussian model is tractable, given the current posterior $p(\lambda_{t-1}^{\text{VB}}|\hat{\lambda}_{1:t-1}) = \mathcal{N}(\mu_{t-1}; \Sigma_{t-1})$ and a noisy coordinate update $\hat{\lambda}_t$, the next posterior is computed directly using Gaussian manipulations [11],

$$p(\lambda_t^{\text{VB}}|\hat{\lambda}_{1:t}) = \mathcal{N}\left([1 - \mathrm{P}_t]\mu_{t-1} + \mathrm{P}_t\hat{\lambda}_t, [1 - \mathrm{P}_t]^{-1}[\Sigma_{t-1} + Q]\right), \tag{7}$$

$$\mathrm{P}_t = [\Sigma_{t-1} + Q][\Sigma_{t-1} + Q + R]^{-1}. \tag{8}$$

The variable $\mathrm{P}_t$ is known as the Kalman gain. Notice the update to the posterior mean has the same form as the SVI update in Eq 1. The gain $\mathrm{P}_t$ is directly equivalent to the SVI step size $\rho_t$.[3] Different modelling choices to get different optimization schedules. We now present some key cases.

**Static Parameters** If the parameters $Q$ and $R$ are fixed, the step size progression in Eq 7 can be computed *a priori* as $\mathrm{P}_{t+1} = [Q/R + \mathrm{P}_t][1 + Q/R + \mathrm{P}_t]^{-1}$. This yields a fixed sequence of decreasing step size. A popular schedule is the Robbins-Monro routine, $\rho \propto (t_0 + t)^{-\kappa}$ also used in SVI [1]. If we set $Q = 0$ the variational filter returns a Robbins-Monro schedule with $\kappa = 1$. This corresponds to online estimation of the mean of a Gaussian. This is because $Q = 0$ assumes that the optimization has converged and the filter simply averages the noisy updates.

In practice, decay rates slower that $\kappa = 1$ perform better [2, 8]. This is because updates which were computed using old parameter values are forgotten faster. Setting $Q > 0$ yields the same reduced memory. In this case, the step size tends to a constant $\lim_{t\to\infty} \mathrm{P}_t = [\sqrt{1 + 4R/Q} + 1][\sqrt{1 + 4R/Q} + 1 + 2R/Q]^{-1}$. Larger the noise-to-signal ratios $R/Q$ result in smaller limiting step sizes. This demonstrates the automatic bias/variance trade-off. If $R/Q$ is large, the variance in the noisy updates $\mathrm{Var}[\hat{\lambda}_t]$ is assumed large. Therefore, the filter uses a smaller step size, yielding more bias (Eq 5), but with lower overall error. Conversely, if there is no noise $R/Q = 0$, $\mathrm{P}_\infty = 1$ and we recover batch VB.

**Parameter Estimation** Normally the parameters will not be known *a priori*. Further, if $Q$ is fixed then the step size does not tend to zero and so Robbins-Monro criteria do not hold [7]. We can address both issues by estimating $Q$ and $R$ online.

The parameter $R$ models the variance in the noisy optima, and $Q$ measures how near the process is to convergence. These parameters are unknown and will change as the optimization progresses. $Q$ will decrease as convergence is approached; $R$ may decrease or increase. In our demonstration in Fig. 1, it increases during early iterations and then plateaus. Therefore we estimate these parameters online, similar to [8, 12]. The desired parameter values are

$$R = \mathbb{E}[||\hat{\lambda}_t - \lambda_t^{\text{VB}}||^2] = \mathbb{E}[||\hat{\lambda}_t - \lambda_{t-1}^{\text{VB}}||_2^2] - ||\lambda_t^{\text{VB}} - \lambda_{t-1}^{\text{VB}}||_2^2, \tag{9}$$

$$Q = ||\lambda_t^{\text{VB}} - \lambda_{t-1}^{\text{VB}}||_2^2. \tag{10}$$

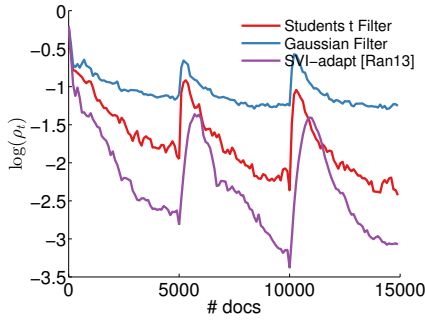

Figure 2: Step sizes learned by the Gaussian Kalman filter, the Student's t filter (Alg. 1) and the adaptive learning rate in [8], on non-stationary ArXiv data. The adaptive algorithms react to the dataset shift by increasing the step size. The variational filters react even faster than adaptive-SVI because not only do $Q$ and $R$ adjust, but the posterior variance increases at the shift which further augments the next step size.

We estimate these using exponentially weighted moving averages. To estimate the two terms in Eq 9, we estimate the expected difference between the current state and the observation $g_t = \mathbb{E}[\hat{\lambda}_t - \lambda_{t-1}^{\text{VB}}]$, and the norm of this difference $h_t = \mathbb{E}[||\hat{\lambda}_t - \lambda_{t-1}^{\text{VB}}||_2^2]$, using

$$g_t = (1 - \tau_t^{-1})g_{t-1} + \tau_t^{-1}(\hat{\lambda}_t - \mu_{t-1}), \quad h_t = (1 - \tau_t^{-1})h_{t-1} + \tau_t^{-1}||\hat{\lambda}_t - \mu_{t-1}||_2^2, \quad (11)$$

where $\tau$ is the window length and $\mu_{t-1}$ is the current posterior mean. The parameters are estimated as $R = h_t - ||g_t||_2^2$ and $Q = ||g_t||_2^2$. After filtering, the window length is adjusted to $\tau_{t+1} = (1 - \text{P}_t)\tau_t + 1$. Larger steps result in shorter memory of old parameter values. Joint parameter and state estimation can be poorly determined. Initializing the parameters to appropriate values with Monte Carlo sampling, as in [8], mitigates this issue. In our experiments we avoid this underspecification by tying the filtering parameters across the filters for each variational parameter.

The variational filter with parameter estimation recovers an automatic step size similar to the adaptive-SVI algorithm in [8]. Their step size is equivalent to $\rho_t = Q/[Q + R]$. Variational filtering uses $\text{P}_t = [\Sigma_{t-1} + Q]/[\Sigma_{t-1} + Q + R]$, Eq 7. If this posterior variance $\Sigma_{t-1}$ is zero the updates are identical. If $\Sigma_{t-1}$ is large, as in early time steps, the filter produces a larger step size. Fig. 3 demonstrates how the these methods react to non-stationary data. LDA was run on ArXiv abstracts whose category changed every 5k documents. Variational filtering and adaptive-SVI react to the shift by increasing the step size, the ELBO is similar for both methods.

**Student's t Filter** In SVI, the noisy estimates $\hat{\lambda}_t$ are often heavy-tailed. For example, in matrix factorization heavy-tailed parameters distributions [13] produce to heavy-tailed noisy updates. Empirically, we observe similar heavy tails in LDA. Heavy tails may also arise from computing Euclidean distances between parameter vectors and not using the more natural Fisher information metric [9]. We add robustness these sources of noise with a heavy-tailed Kalman filter.

We use a t-distributed noise model, $p(\hat{\lambda}_t|\lambda_t^{\text{VB}}) = \mathcal{T}(\lambda_t^{\text{VB}}, R, \delta)$, where $\mathcal{T}(m, V, d)$ denotes a t-distribution with mean $m$, covariance $V$ and $d$ degrees of freedom. For computational convenience we also use a t-distributed transition model, $p(\lambda_{t+1}^{\text{VB}}|\lambda_t^{\text{VB}}) = \mathcal{T}(\lambda_t^{\text{VB}}, Q, \gamma)$. If the current posterior is t-distributed, $p(\lambda_t^{\text{VB}}|\hat{\lambda}_{1:t}) = \mathcal{T}(\mu_t, \Sigma_t, \eta_t)$ and the degrees of freedom are identical, $\eta_t = \gamma = \delta$, then filtering has closed-form,

$$p(\lambda_t^{\text{VB}}|\hat{\lambda}_{1:t}) = \mathcal{T}\left((1 - \text{P}_t)\mu_{t-1} + \text{P}_t\hat{\lambda}_t, \frac{\eta_{t-1} + \Delta^2}{\eta_{t-1} + ||\lambda||_0}(1 - \text{P}_t)[\Sigma_{t-1} + Q], \eta_{t-1} + ||\lambda||_0\right), \quad (12)$$

$$\text{where } \text{P}_t = \frac{\Sigma_{t-1} + Q}{\Sigma_{t-1} + Q + R}, \text{ and } \Delta^2 = \frac{||\hat{\lambda}_t - \mu_{t-1}||_2^2}{\Sigma_{t-1} + Q + R}. \quad (13)$$

The update to the mean is the same as in the Gaussian KF. The crucial difference is in the update to the variance in Eq 12. If an outlier $\hat{\lambda}_t$ arrives, then $\Delta^2$, and hence $\Sigma_t$, are augmented. The increased posterior uncertainty at time $t + 1$ yields an increased gain $\text{P}_{t+1}$. This allows the filter to react quickly to a large perturbation. The t-filter differs fundamentally to the Gaussian KF in that the step size is now a direct function of the observations. In the Gaussian KF the dependency is indirect, through the estimation of $R$ and $Q$.

Eq 12 has closed-form because the d.o.f. are equal. Unfortunately, this will not generally be the case because the posterior degrees of freedom grow, so we require an approximation. Following [14], we approximate the 'incompatible' t-distributions by adjusting their degrees of freedom to be equal. We choose all of these to equal $\tilde{\eta}_t = \min(\eta_t, \gamma, \delta)$. We match the degrees of freedom in

this way because it prevents the posterior degree of freedom from growing over time. If $\eta_t$, Eq 12 were allowed to grow large, the t-distributed filter reverts back to a Gaussian KF. This is undesirable because the heavy-tailed noise does not necessarily disappear at convergence.

To account for adjusting the degrees of freedom, we moment match the old and new t-distributions. This has closed-from; to match the second moments of $\mathcal{T}(m, \tilde{\Sigma}, \tilde{\eta})$ to $\mathcal{T}(m, \Sigma, \eta)$, the variance is set to $\tilde{\Sigma} = \frac{\eta(\tilde{\eta}-2)}{(\eta-2)\tilde{\eta}}\Sigma$. This results in tractable filtering and has the same computational cost as Gaussian filtering. The routine is summarized in Algorithm 1.

---

**Algorithm 1** Variational filtering with Student's t-distributed noise

---

1: **procedure** FILTER(data $x_{1:N}$)
2:     Initialize filtering distribution $\Sigma_0$, $\mu_0$, $\eta_0$, see § 5
3:     Initialize statistics $g_0, h_0, \tau_0$ with Monte-Carlo sampling
4:     Set initial variational parameters $\lambda_0 \leftarrow \mu_0$
5:     **for** $t = 1, \dots, T$ **do**
6:         Sample a datapoint $x_t$                              ▷ Or a mini-batch of data.
7:         $\hat{\lambda}_t \leftarrow f(\lambda_t, x_t)$, $f$ given by Equation Eq 4 ▷ Noisy estimate of the coordinate optimum.
8:         Compute $g_t$ and $h_t$ using Eq 11.
9:         $R \leftarrow h_t - g_t^2, \quad Q \leftarrow h_t$                     ▷ Update parameters of the filter.
10:        $\tilde{\eta}_{t-1} \leftarrow \min(\eta_{t-1}, \gamma, \delta)$                    ▷ Match degrees of freedom.
11:        $\tilde{\Sigma}_{t-1} \leftarrow \eta_{t-1}(\tilde{\eta}_{t-1} - 2)[(\eta_{t-1} - 2)\tilde{\eta}_{t-1}]^{-1}\Sigma_{t-1}$, similar for $\tilde{R}, \tilde{Q}$   ▷ Moment match.
12:        $P_t \leftarrow [\tilde{\Sigma}_{t-1} + \tilde{Q}][\tilde{\Sigma}_{t-1} + \tilde{Q} + \tilde{R}]^{-1}$          ▷ Compute gain, or step size.
13:        $\Delta^2 \leftarrow ||\hat{\lambda}_t - \mu_{t-1}||_2^2[\tilde{\Sigma}_{t-1} + \tilde{Q} + \tilde{R}]^{-1}$
14:        $\mu_t \leftarrow [I - P_t]\mu_{t-1} + P_t\hat{\lambda}_t,$                  ▷ Update filter posterior.
15:        $\Sigma_t \leftarrow \frac{\tilde{\eta}_{t-1} + \Delta^2}{\tilde{\eta}_{t-1} + ||\lambda||_0}[I - P_t][\tilde{\Sigma}_{t-1} + \tilde{Q}], \quad \eta_t \leftarrow \eta_{t-1} + 1$
16:        $\lambda_t \leftarrow \mu_t$                           ▷ Update the variational parameters of $q$.
17:     **end for**
18:     **return** $\lambda_T$
19: **end procedure**

---

## 4 Related Work

**Stochastic and Streamed VB**    SVI performs fast inference on a fixed dataset of known size $N$. Online VB algorithms process an infinite stream of data [15, 16], but these methods cannot use a re-sampled datapoint. Variational filtering falls between both camps. The noisy observations require an estimate of $N$. However, Kalman filtering does not try to optimize a static dataset like a fixed Robbins-Monro schedule. As observed in Fig. 3 the algorithm can adapt to a regime change, and forgets the old data. The filter simply tries to move to the VB coordinate update at each step, and is not directly concerned about asymptotic convergence on static dataset.

**Kalman filters for parameter learning**    Kalman filters have been used to learn neural network parameters. Extended Kalman filters have been used to train supervised networks [17, 18, 19]. The network weights evolve because of data non-stationarity. This problem differs fundamentally to SVI. In the neural network setting, the observations are the fixed data labels, but in SVI the observations are noisy realizations of a moving VB parallel coordinate optimum. If the VF draws the same datapoint, the observations $\hat{\lambda}$ will still change because $\lambda_t$ will have changed. In the work with neural nets, the same datapoint always yields the same observation for the filter.

**Adaptive learning rates**    Automatic step size schedules have been proposed for online estimation of the mean of a Gaussian [20], or drifting parameters [21]. The latter work uses a Gaussian KF for parameter estimation in approximate dynamic programming. Automatic step sizes are derived for stochastic gradient descent in [12] and SVI in [8]. These methods set the step size to minimize the expected update error. Our work is the first Bayesian approach to learn the SVI schedule.

**Meta-modelling**    Variational filtering is a 'meta-model', these are models that assist training of a more complex method. They are becoming increasingly popular, examples include Kalman filters

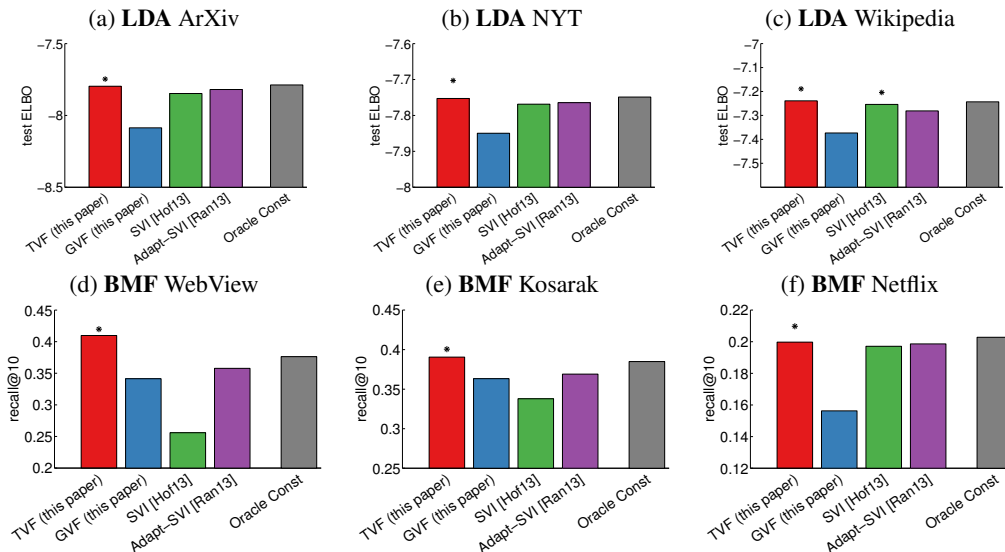

Figure 3: Final performance achieved by each algorithm on the two problems. Stars indicate the best performing non-oracle algorithm and those statistically indistinguishable at $p = 0.05$. (a-c) LDA: Value of the ELBO after observing $0.5M$ documents. (d-f) BMF: recall@10 after observing $2 \cdot 10^8$ cells.

for training neural networks [17], Gaussian process optimization for hyperparameter search [22] and Gaussian process regression to construct Bayesian quasi-Newton methods [23].

## 5 Empirical Case Studies

We tested variational filtering on two diverse problems: topic modelling with Latent Dirichlet Allocation (LDA) [24], a popular testbed for scalable inference routines, and binary matrix factorization (BMF). Variational filtering outperforms Robbins-Monro SVI and a state-of-the-art adaptive method [8] in both domains. The Student's t filter performs substantially better than the Gaussian KF and is competitive with an oracle that picks the best constant step size with hindsight.

**Models** We used 100 topics in LDA and set the Dirichlet hyperparameters to $0.5$. This value is slightly larger than usual because it helps the stochastic routines escape local minima early on. For BMF we used a logistic matrix factorization model with a Gaussian variational posterior over the latent matrices [3]. This task differs to LDA in two ways. The variational parameters are Gaussian and we sample single cells from the matrix to form stochastic updates. We used minibatches of 100 documents in LDA, and 5 times the number of rows in BMF.

**Datasets** We trained LDA on three large document corpora: 630k abstracts from the ArXiv, 1.73M New York Times articles, and Wikipedia, which has $\approx 4M$ articles. For BMF we used three recommendation matrices: clickstream data from the Kosarak news portal; click data from an e-commerce website, BMS-WebView-2 [25]; and the Netflix data, treating 4-5 star ratings as ones. Following [3] we kept the 1000 items with most ones and sampled up to 40k users.

**Algorithms** We ran our Student's t variational filter in Algorithm 1 (TVF) and the Gaussian version in § 3 (GVF). The variational parameters were initialized randomly in LDA and with an SVD-based routine [26] in BMF. The prior variance was set to $\Sigma_0 = 10^3$ and t-distribution's degrees of freedom to $\eta_0 = 3$ to get the heaviest tails with a finite variance for moment matching.

In general, VF can learn full-rank matrix stepsizes. LDA and BMF, however, have many parameters, and so we used the simplest setting of VF in which a single step size was learned for all of them; that is, $Q$ and $R$ are constrained to be proportional to the identity matrix. This choice reduces the cost of VF from $\mathcal{O}(N^3)$ to $\mathcal{O}(N)$. Empirically, this computational overhead was negligible. Also

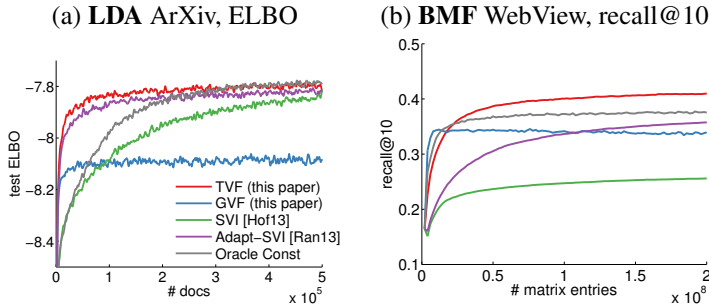

(a) **LDA** ArXiv, ELBO  (b) **BMF** WebView, recall@10

Figure 4: Example learning curves of (a) the ELBO (plot smoothed with Lowess' method) and (b) recall@10, on the LDA and BMF problems, respectively.

it allows us to aggregate statistics across the variational parameters, yielding more robust estimates. Finally, we can directly compare our Bayesian adaptive rate to the single adaptive rate in [8].

We compared to the SVI schedule proposed in [1]. This is a Robbins-Monro schedule $\rho_t = (t_0 + t)^{-\kappa}$, we used $\kappa = 0.7; t_0 = 1000$ for LDA as these performed well in [1, 2, 8] and $\kappa = 0.7, t_0 = 0$ for BMF, as in [3]. We also compared to the adaptive-SVI routine in [8]. Finally, we used an oracle method that picked the constant learning rate from a grid of rates $10^{-k}, k \in 1, \dots, 5$, that gave the best final performance. In BMF, the Robbins-Monro SVI schedule learns a different rate for each row and column. All other methods computed a single rate.

**Evaluation**   In LDA, we evaluated the algorithms using the per-word ELBO, estimated on random sets of held-out documents. Each algorithm was given $0.5M$ documents and the final ELBO was averaged over the final 10% of the iterations. We computed statistical significance between the algorithms with a t-test on these noisy estimates of the ELBO. Our BMF datasets were from item recommendation problems, for which recall is a popular metric [27]. We computed recall at $N$ by removing a single one from each row during training. We then ranked the zeros by their posterior probability of being a one and computed the fraction of the rows in which the held-out one was in the top $N$. We used a budget of $2 \cdot 10^8$ observations and computed statistical significance over 8 repeats of the experiment, including the random train/test split.

**Results**   The final performance levels on both tasks are plotted in Fig. 3. These plots show that over the six datasets and two tasks the Student's t variational filter is the strongest non-oracle method. SVI [1] and Adapt-SVI [8] come close on LDA, which they were originally used for, but on the WebView and Kosarak binary matrices they yield a substantially lower recall. In terms of the ELBO in BMF (not plotted), TVF was the best non-oracle method on WebView and Kosarak and SVI was best on Netflix, with TVF second best. The Gaussian Kalman filter worked less well. It produced high learning rates due to the inaccurate Gaussian noise assumption.

The t-distributed filter appears to be robust to highly non-Gaussian noise. It was even competitive with the oracle method (2 wins, 2 draws, 1 loss). Note that the oracle picked the best final performance at time $T$, but at $t < T$ the variational filter converged faster, particularly in LDA. Fig. 4 (a) shows example learning curves on the ArXiv data. Although the oracle just outperforms TVF at $0.5M$ documents, TVF converged much faster. Fig. 4 (b) shows example learning curves in BMF on the WebView data. This figure shows that most of the BMF routines converge within the budget. Again, TVF not only reached the best solution, but also converged fastest.

**Conclusions**   We have presented a new perspective on SVI as approximate parallel coordinate descent. With our model-based approach to this problem, we shift the requirement from hand tuning optimization schedules to constructing an appropriate tracking model. This approach allows us to derive a new algorithm for robust SVI that uses a model with Student's t-distributed noise. This Student's t variational filtering algorithm performed strongly on two domains with completely different variational distributions. Variational filtering is a promising new direction for SVI.

**Acknowledgements**   NMTH is grateful to the Google European Doctoral Fellowship scheme for funding this research. DMB is supported by NSF CAREER NSF IIS-0745520, NSF BIGDATA NSF IIS-1247664, NSF NEURO NSF IIS-1009542, ONR N00014-11-1-0651 and DARPA FA8750-14-2-0009. We thank James McInerney, Alp Kucukelbir, Stephan Mandt, Rajesh Ranganath, Maxim Rabinovich, David Duvenaud, Thang Bui and the anonymous reviews for insightful feedback.

## Footnotes

*Work carried out while a member of the University of Cambridge, visiting Princeton University.

[1]To readers familiar with stochastic variational inference, we refer to the global variational parameters, assuming that the local parameters are optimized at each iteration. Details can be found in [1].

[2] We do not perform 'smoothing' in our dynamical system because we are not interested in old VB coordinate optima after the parameters have been optimized further.

[3] In general, $\mathrm{P}_t$ is a full-rank matrix update. For simplicity, and to compare to scalar learning rates, we present the 1D case. The multi-dimensional generalization is straightforward.

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
