[Reviews · NeurIPS 2014]

Submitted by Assigned_Reviewer_9

Stochastic variational inference (SVI) requires careful selection of a step size. This paper proposes a Kalman filter to set the step size automatically. The authors show that standard Gaussian KF does not satisfy the Robbins Munro criteria (and performs badly). They propose to apply a KF based on T-distributions, and show that this gives better results than standard SVI.

The problem addressed in this paper is relevant. The solutions proposed are also quite interesting and seem useful. I have few theoretical and computational concerns which I have discussed below. The paper is clearly written. The work is original and significant. The numerical experiments shown are satisfactory.

Below are some of my comments to help improve the quality of this paper.

Eq. 7 and 8 involve inverting a matrix at every iteration of the algorithm. Since Q and R change every time, we cannot precompute the step sizes. Does this increase the computation significantly? I would expect that there is some increase due to inversion of the matrices who size depends on size of lambda. Will you please discuss the extent of computation increase you notice in your experiments?

Doing filtering and parameter estimation at the same time for Kalman filter should be problematic, at least initially. I don't see that in the Fig. 4. Why is that? Do you use some method to compensate for this? In any case, it will be good to include a discussion about the consequences of this approach. When is it going to work and when is it going to be troublesome? Does the size of lambda play a role in this too? etc.

You mention that for fixed Q and R Robbins-Munro criteria do not hold. When they are varying, does it holds for sure? can you prove it? Is it the same for T-distribution KF? Is this the reason why in Fig. 4 Gaussian KF doesn't reach the true maximum?

There are small typos here and there. There is a mistake in Eq. 5. The first equation works for t, while the second equation works with t+1.
Summary: The problem addressed in this paper is relevant. The solutions proposed are also interesting and useful.

Submitted by Assigned_Reviewer_36

This paper combines Bayesian filtering with stochastic variational inference (SVI), which improves the estimation of the batch variational Bayes (VB) from its noisy estimate.

Although the approach is not technically so deep, its usefulness is demonstrated through empirical studies.

p.5, l.258: The extension to Student's t filter is interesting. However, I wonder what the clause "If the posterior is also t-distributed" means. Does the Bayes rule determine the posterior from p(\lambda_{t+1} | \lambda_t) and p(\hat{\lambda}_t | VB(\lambda_t)) (This notation is not consistent with Eq.(6))?

The author feedback clarified this point.

minor:
p.2, l.83: [q||p] -> KL[q||p]
p.5, Eq.(10): VB(\lambda_t) -> VB(\lambda_{t-1})
p.8, l.423: demonstrates show that -> demonstrates that
p.8, l.424: but we it also -> but it also
Summary: The proposed approach is useful for large data sets while it is basically a simple application of filtering methods to tracking the variational Bayes posterior.

Submitted by Assigned_Reviewer_41

This paper presented a framework to model the adaptive step-size of the stochastic variational inference (SVI) by using Kalman filter.
Although the adaptive SVI has already proposed, this work is the first Bayesian approach to learn scheduling step-size of the SVI, which is an interesting direction of SVI.

The main concerns are the computational cost about the dimension and the convergence of the proposed algorithm.
It is well known that Kalman filter doesn't work well in the large dimension.
You used 100 topics in LDA, which is too small for the large scale documents.
We typically used more than 1000 topics in the large scale documents.
I'm interested in the total computation complexity of your framework.

The convergence of the algorithm is important in online learning.
Surely, it has not been developed in the existing adaptive SVI but they showed the some conditions.
Would you show the necessary conditions of the convergence of your algorithm and their reasonability in real applications.

Summary: This paper proposed the Bayesian approach for the adaptive step-size of the stochastic variational inference.
The main concerns are the computational cost about the dimension and the convergence of the proposed algorithm.

Submitted by Assigned_Reviewer_46

The paper starts with the previously proposed stochastic variational inference which updates parameters using small batches, but changes the new values of variational parameters to a linear combination of the old ones and the new ones, with some suitable learning rate that controls this combination. Then, the paper proposes that this is essentially temporal filtering, and then proposes to do this with an adaptive, rather than a trivial filter. The math in the paper is a bit unclear - are Q and R matrices in any of their implementations, or are they just scalars? In other words, is some correlation across parameters in the vector lambda being tracked, or is the whole thing simply a way to avoid setting the scalar learning rate (and its decay schedule) by hand? In the former case, the authors should have discussed what was learned in these matrices.

I will adjust my grade based on the clarifications in the rebuttal.

In terms of experimental results, the paper does show improvements over other SVI methods for particular setting of the model (100 LDA topics), but it does not go further into evaluating if simply increasing the topics would lead to improvements in recall rates and if for best model complexity the new method still ahs advantages over SVI, and if it has advantages over other learning techniques.

As it stands, I find the paper potentially very interesting, but it seems to stop a little short from the goal.
Summary: see above
Author Feedback
Author rebuttal: Thank-you very much for your thoughtful comments. We want to re-emphasize two substantial practical impacts of our work in automatically learning the step-size for SVI:

1) Our t-distributed filter has strong empirical performance on two diverse domains. Algorithm 1 can be applied immediately by practitioners.
2) Our new Bayesian framework for SVI opens the door for any filtering models to be used. Assumptions about the learning process are now easy to encode and are automatically incorporated into the learning rate. Only with this framework could we design the strong performing t-filter.

R41, R46, R9 raise questions about the filtering parameters Q and R, and computational complexity. We agree this is not fully explained. We shall add the following clarification:

With N parameters the filter can learn any matrix-variate learning rate, and the cost is O(N^3). In our equations and experiments we treat each parameter as an independent unidimensional filter (see footnote 3 p4, and lines 391-39, p8). Equivalently, we restrict Q and R to be scalar multiples of the identity matrix. The reasons for this are:

1) The models we investigated, LDA and BMF, have many parameters and the stochastic updates are sparse (documents contain only a small fraction of the vocab, and most matrix entries are zeros). By tying Q and R across the filters we share statistics, leading to robust estimation of these parameters.
2) The complexity of the filter is reduced to O(N), and the additional cost is negligible.
3) We learn a single learning rate as in current algorithms [1,8,22]. This provides a fair comparison that indicates that our variational filter learns a better rate, and it is not the use of parameter-dependent or correlated rates providing better performance.

Learning matrix-variate rates would be an interesting extension, we speculate that this would be most useful for models with few parameters. We are pleased to achieve state-of-the-art performance in the simpler scalar setting.

Specific comments:

R36:

>p.5, l.258: ... Does the Bayes rule determine the posterior from p(\lambda_{t+1} | \lambda_t) and p(\hat{\lambda}_t | VB(\lambda_t)) (This notation is not consistent with Eq.(6))?

We shall add clarification: If the *current* posterior p(\lambda_t | \hat{\lambda}_{1:t}) is t-distributed and \eta=\gamma=\delta, then Bayes rule can be computed in closed form: p(\lambda_{t+1} | \hat{\lambda}_{1:t+1}) \propto p(\hat{\lambda}_{t+1} | \lambda_{t+1}) p(\lambda_t | \hat{\lambda}_{1:t}), and the posterior remains t-distributed. Since this is rarely the case (l.274) we use approximations. Line 255 should read p(\hat{\lambda_t}|\lambda_t).

R41:

See comment at top for computational cost.

> You used 100 topics in LDA, which is too small...

We used K=100 topics for consistency with the experiments in [1,2,8]. The relative performances of the algorithms are the same for different K and for different latent dimensionalities in BMF. We shall add a figure to confirm.

>Surely, it has not been developed in the existing adaptive SVI but they showed the some conditions.
>Would you show the necessary conditions of the convergence of your algorithm and their reasonability in real applications.

In our experiments the algorithms converged without fail. Proving convergence in the simpler, non-Bayesian, adaptive learning rate in [8] is an open problem. As noted in [8], proving convergence to the optimal batch update in SVI is hard becuase it moves at each iteration. They only show convergence under an idealised, but uncomputable, learning rate to a static local optimum.

Although theoretically appealing, asymptotic convergence guarantees are not the primary focus of our research. This is because with large or streamed datasets, where the data is often non-stationary, the algorithm is unlikely to achieve full convergence in practice. In these settings, issues of convergence are less meaningful. Our method makes a Bayesian bias/variance trade-off to optimize the step at each time step, and hence is robust to non-stationary data (see Fig. 2).

R46:

See comment at top regarding the Q and R parameters.

> the paper does show improvements over other SVI methods for particular setting of the model ... if for best model complexity the new method still has advantages over SVI

Our goal is to solve the optimization problem in SVI. For a fair test of this ability, we set the model complexity to be the same for each algorithm. Each algorithm will converge eventually to the same optimum but at different rates (modulo issues of local minima). With different model complexities we would confound the convergence rates produced by each algorithm with different optimization surfaces. Our experimental setup is consistent with previous work on SVI [1,2,8]. With more topics the relative performances are the same.

R9:

See comment at top regarding matrix inversions and cost.

>Doing filtering and parameter estimation at the same time for Kalman filter should be problematic ... I don't see that in the Fig. 4. Why ...

We have no problem because we share filtering parameters (Q, R) across the filters (see comments at top). If a higher rank Q and R are used they could be initialised with Monte Carlo as in [8]. We shall add this discussion.

>You mention that for fixed Q and R Robbins-Munro criteria do not hold. When they are varying, does it holds for sure? ... Is this the reason why in Fig. 4 Gaussian KF doesn't reach the true maximum?

The R-M criteria do not hold in general for variational filtering, nor do they hold for the adaptive learning rate [8]. See comments to R41.

Most theory for SGD applies to convex problems, which is usually not the case in SVI. In Fig. 4 GKF has converged to a worse *local* optimum due to too making too large steps early on in learning. The performance difference is not due to R-M asymptotic convergence issues.